

# A new species of *Nanhsiungchelys* (Testudines: Cryptodira: Nanhsiungchelyidae) from the Upper Cretaceous of Nanxiong Basin, China

Yuzheng Ke[1], Imran A. Rahman[2,3], Hanchen Song[1], Jinfeng Hu[1], Kecheng Niu[4,5], Fasheng Lou[6], Hongwei Li[7] and Fenglu Han[1]

[1] School of Earth Science, China University of Geosciences (Wuhan), Wuhan, Hubei, People's Republic of China
[2] The Natural History Museum, London, United Kingdom
[3] Oxford University Museum of Natural History, Oxford, United Kingdom
[4] State Key Laboratory of Cellular Stress Biology, School of Life Sciences, Xiamen University, Xiamen, Fujian, People's Republic of China
[5] Yingliang Stone Natural History Museum, Nan'an, Fujian, People's Republic of China
[6] Jiangxi Geological Survey and Exploration Institute, Nanchang, Jiangxi, People's Republic of China
[7] Guangdong Geological Survey Institute, Guangzhou, Guangdong, People's Republic of China

Corresponding authors
Yuzheng Ke, key1480@163.com
Fenglu Han, hanfl@cug.edu.cn

## ABSTRACT

Nanhsiungchelyidae are a group of large turtles that lived in Asia and North America during the Cretaceous. Here we report a new species of nanhsiungchelyid, *Nanhsiungchelys yangi* sp. nov., from the Upper Cretaceous of Nanxiong Basin, China. The specimen consists of a well-preserved skull and lower jaw, as well as the anterior parts of the carapace and plastron. The diagnostic features of *Nanhsiungchelys* include a large entire carapace length (∼55.5 cm), a network of sculptures consisting of pits and ridges on the surface of the skull and shell, shallow cheek emargination and temporal emargination, deep nuchal emargination, and a pair of anterolateral processes on the carapace. However, *Nanhsiungchelys yangi* differs from the other species of *Nanhsiungchelys* mainly in having a triangular-shaped snout (in dorsal view) and wide anterolateral processes on the carapace. Additionally, some other characteristics (*e.g.*, the premaxilla is higher than wide, the maxilla is unseen in dorsal views, a small portion of the maxilla extends posterior and ventral of the orbit, and the parietal is bigger than the frontal) are strong evidence to distinguish *Nanhsiungchelys yangi* from *Nanhsiungchelys wuchingensis*. A phylogenetic analysis of nanhsiungchelyids places *Nanhsiungchelys yangi* and *Nanhsiungchelys wuchingensis* as sister taxa. *Nanhsiungchelys yangi* and some other nanhsiungchelyids bear distinct anterolateral processes on the carapace, which have not been reported in any extant turtles and may have played a role in protecting the head. The Nanxiong Basin was extremely hot during the Late Cretaceous, and so we suggest that nanhsiungchelyids might have immersed themselves in mud or water to avoid the heat, similar to some extant tortoises. If they were capable of swimming, our computer simulations of fluid flow suggest the anterolateral processes could have reduced drag during locomotion.

## INTRODUCTION

Nanhsiungchelyidae are an extinct group of Pan-Trionychia, which lived in Asia and North America from the Early Cretaceous until their extinction at the Cretaceous–Paleogene boundary (*Hirayama, Brinkman & Danilov, 2000*; *Li & Tong, 2017*; *Joyce et al., 2021*). These turtles are characterized by a large body size (maximum carapace length of about 120 cm as preserved), flat carapace relative to tortoises, stubby elephantine limbs, and shells covered with a network of sculptures consisting of pits and ridges (*Yeh, 1966*; *Hutchison & Archibald, 1986*; *Brinkman et al., 2015*; *Hu et al., 2016*; *Li & Tong, 2017*). In addition, these turtles produced thick-shelled (∼1.8 mm) eggs and are thought to have had similar reproductive strategies to extant tortoises (*e.g.*, large and spherical eggs) (*Ke et al., 2021*). Recently, the morphology and phylogenetic relationships of nanhsiungchelyids have been studied in detail (*Danilov, Sukhanov & Syromyatnikova, 2013*; *Brinkman et al., 2015*; *Tong et al., 2016*; *Mallon & Brinkman, 2018*; *Tong & Li, 2019*). Among the eight genera of Nanhsiungchelyidae, most taxa typically have a relatively short carapace, shallow nuchal emargination, narrow neurals and vertebral scutes, and lack large anterior processes on the carapace (*Tong & Li, 2019*). In contrast, *Nanhsiungchelys* and *Anomalochelys* (which form a sister group) share an elongated shell, a wide and deep nuchal emargination, large anterior processes on the carapace, wide neurals and vertebral scutes, and a sub-triangular first vertebral scute with a very narrow anterior end (*Tong & Li, 2019*). These two genera have only been found in southern China and Japan (*Hirayama et al., 2001*; *Hirayama et al., 2009*; *Li & Tong, 2017*; *Tong & Li, 2019*), whereas other nanhsiungchelyids have a wider geographical distribution (*Danilov & Syromyatnikova, 2008*; *Mallon & Brinkman, 2018*).

*Nanhsiungchelys* and *Anomalochelys* are unique among Mesozoic turtles in possessing distinct anterolateral processes on the carapace, with a similar body structure known in the Miocene side-necked turtle *Stupendemys geographicus* (*Cadena et al., 2020*). Palaeontologists have debated whether nanhsiungchelyids were aquatic or terrestrial for nearly 60 years (see *Mallon & Brinkman (2018)* for a detailed overview), but the ecological role of the anterolateral processes has largely been ignored. It was previously suggested they played a role in protecting the head (*Hirayama et al., 2001*) or facilitating sexual displays (*Hirayama & Sonoda, 2012*), but further study of their function is required.

In China, six species of nanhsiungchelyids have been reported (Table 1), with many specimens recovered from the Upper Cretaceous of Nanxiong Basin, Guangdong Province. *Yeh (1966)* described the first species, *Nanhsiungchelys wuchingensis*, which was restudied by *Tong & Li (2019)*. *Hirayama et al. (2009)* provided a preliminary study of a large Cretaceous turtle (SNHM 1558) which they placed within Nanhsiungchelyidae; *Li & Tong (2017)* later attributed this to *Nanhsiungchelys*. In addition, two eggs (IVPP V2789) from Nanxiong Basin were assigned to nanhsiungchelyids based on their co-occurrence with *Nanhsiungchelys wuchingensis (Young, 1965)*.

Nanxiong Basin (Fig. 1A) is a NE-trending faulted basin controlled by the Nanxiong Fault in the northern margin, covering an area of about 1,800 km$^2$ and spanning Guangdong and Jiangxi provinces in China (*Zhang et al., 2013*). Well-exposed outcrops of Cretaceous–Paleogene strata occur in Nanxiong Basin (*Ling, Zhang & Lin, 2005*), and

**Table 1 Taxonomy and distribution of Nanhsiungchelyidae in China.**

| Taxa | Specimen Number | Location | Age | Stratigraphic Unit | References |
|---|---|---|---|---|---|
| *Nanhsiungchelys wuchingensis* | IVPP V3106 | Nanxiong, Guangdong | Late Cretaceous (Cenomanian–middle Campanian) | Dafeng Formation | *Yeh (1966)* *Tong & Li (2019)* |
| *Nanhsiungchelys* sp. | SNHM 1558 | Nanxiong, Guangdong | Late Cretaceous (Cenomanian–middle Maastrichtian) | Nanxiong Group | *Hirayama et al. (2009)* *Li & Tong (2017)* |
| *Nanhsiungchelys yangi* sp. nov. | CUGW VH108 | Nanxiong, Guangdong | Late Cretaceous (Cenomanian–middle Campanian) | Dafeng Formation | This article |
| *Jiangxichelys neimongolensis* | IVPP RV96007, IVPP RV96008, IVPP 290690-6 RV 96009, IVPP 020790-4 RV 96010, IVPP 130790-1 RV 96011, IMM 4252, IMM 2802, IMM 96NMBY-I-14, IMM 93NMBY-2 | Bayan Mandahu, Inner Mongolia | Late Cretaceous (Campanian) | Wulansuhai Formation | *Brinkman & Peng (1996)* *Brinkman et al. (2015)* *Li & Tong (2017)* |
| *Jiangxichelys ganzhouensis* | NHMG 010415, JXGZ(2012)-178, JXGZ(2012)-179, JXGZ(2012)-180, JXGZ(2012)-182 | Ganzhou, Jiangxi | Late Cretaceous (Maastrichtian) | Lianhe Formation | *Tong & Mo (2010)* *Tong et al. (2016)* |
| *Yuchelys nanyangensis* | HGM NR09-11-14, CUGW EH051 | Nanyang, Henan | Late Cretaceous (Turonian–middle Campanian) | Gaogou Formation | *Tong et al. (2012)* *Ke et al. (2021)* |
| Nanhsiungchelyidae indet. | Specimen number unknown. The authors named it as 'Hefei specimen' | Jiangxi | Late Cretaceous | Unknown | *Hu et al. (2016)* |

**Notes.**
This table does not include small fragments which have less taxonomic significance.

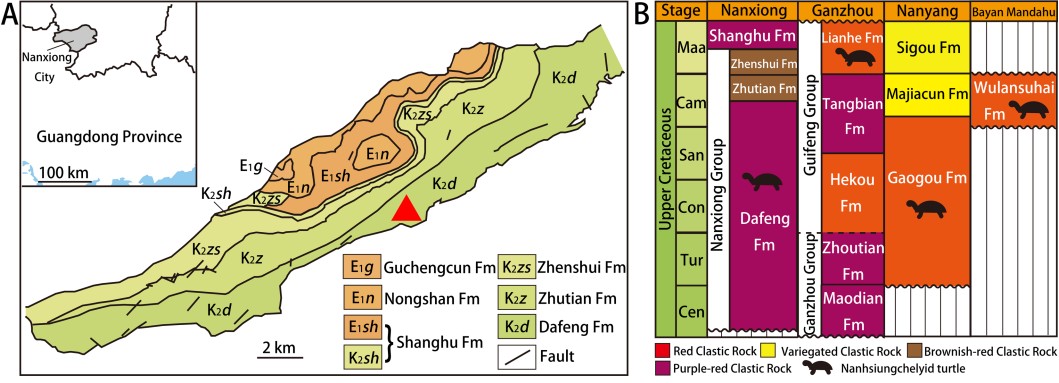

**Figure 1 Geological map of Nanxiong Basin and stratigraphic distribution of valid nanhsiungchelyid turtles in China.** (A) Geological map of Nanxiong Basin, and the red triangle indicates the fossil site, after *Wang et al. (2016)*, *Wang et al. (2019)* and *Xi et al. (2021)*. (B) Stratigraphic distribution of valid nanhsiungchelyid turtles in China. Abbreviations: Cam, Campanian; Cen, Cenomanian; Con, Coniacian; Maa, Maastrichtian; San, Santonian; Tur, Turonian. Stratigraphic information based on work by the *Bureau of Geology and Mineral Exploration and Development of Jiangxi Province (2017)*, *Guangdong Geological Survey Institute (2017)*, *Jerzykiewicz et al. (1993)*, *Xi et al. (2021)*, and *Xu et al. (2015)*.

the lithostratigraphy of the Upper Cretaceous in this region has been studied extensively (see *Zhang et al. (2013)* for details). In 1966, the holotype of *Nanhsiungchelys wuchingensis* (IVPP V3106) was recovered from Nanxiong Basin, with the strata where the fossil was found named the Nanxiong Group (*Yeh, 1966*). Subsequently, *Zhao et al. (1991)* split Nanxiong Group into the upper Pingling Formation and lower Yuanpu Formation, reporting two K–Ar ages for the Yuanpu Formation (67.04 ±2.31 Ma and 67.37 ±1.49 Ma). *Zhang et al. (2013)* further divided the original Yuanpu Formation into the Jiangtou, Yuanpu, Dafeng, and Zhutian formations, with the new Yuanpu Formation just a small part of the original Yuanpu Formation. Most recently, the Yuanpu Formation was eliminated entirely, and the Nanxiong Group now consists of Dafeng, Zhutian, and Zhenshui formations (*Guangdong Geological Survey Institute, 2017*). This terminology was also used by *Xi et al. (2021)*, who summarized lithostratigraphic subdivision and correlation for the Cretaceous of China. According to this scheme, the holotypes of *Nanhsiungchelys wuchingensis* (IVPP V3106) and *N. yangi* (CUGW VH108, see below) both come from the Dafeng Formation.

The Dafeng Formation comprises purple-red, brick-red, and brownish-red conglomerate, sandy conglomerate, and gravel-bearing sandstone, and is intercalated with sandstone, siltstone and silty mudstone (*Guangdong Geological Survey Institute, 2017*). It ranges in age from the Cenomanian to the middle Campanian (*Xi et al., 2021*). In addition to *Nanhsiungchelys*, many vertebrate fossils have been recovered from the Dafeng Formation, including: the dinosaur *Nanshiungosaurus brevispinus* (*Zanno, 2010*); the turtle eggs *Oolithes nanhsiungensis* (*Young, 1965*); and the dinosaur eggs *Macroolithus rugustus*, *Nanhsiungoolithus chuetienensis*, *Ovaloolithus shitangensis*, *O. nanxiongensis*, and *Shixingoolithus erbeni* (*Zhao, Wang & Zhang, 2015*).

Here, we report a new species of *Nanhsiungchelys* from Nanxiong Basin based on a complete skull and partial postcranial skeleton. This allows us to investigate the taxonomy and morphology of nanhsiungchelyids, and based on this we carry out a phylogenetic analysis of the group. In addition, we discuss potential functions of the large anterolateral processes (using computational fluid dynamics to test a possible role in drag reduction) and consider the implications for the ecology of this taxon.

## MATERIALS & METHODS

### Fossil specimen

The specimen (CUGW VH108) consists of a well-preserved skull and lower jaw, together with the anterior parts of the carapace and plastron (Figs. 2–4). This specimen was collected by a local farmer from southeast of Nanxiong Basin, near the Zhenjiang River. Based on the brownish-red siltstone near the skeleton, it was most likely from the Dafeng Formation (*Guangdong Geological Survey Institute, 2017*). CUGW VH108 is housed in the paleontological collections of China University of Geosciences (Wuhan). The skeleton was prepared using an Engraving Pen AT-310 and was photographed with a Canon EOS 6D camera.

### Nomenclatural acts

The electronic version of this article in Portable Document Format (PDF) will represent a published work according to the International Commission on Zoological Nomenclature (ICZN), and hence the new names contained in the electronic version are effectively published under that Code from the electronic edition alone. This published work and the nomenclatural acts it contains have been registered in ZooBank, the online registration system for the ICZN. The ZooBank LSIDs (Life Science Identifiers) can be resolved and the associated information viewed through any standard web browser by appending the LSID to the prefix http://zoobank.org/. The LSID for this publication is: urn:lsid:zoobank.org:pub:F53B5FA5-D018-453D-814D-C854810EFEFE. The online version of this work is archived and available from the following digital repositories: PeerJ, PubMed Central SCIE and CLOCKSS.

### Phylogenetic analysis

Parsimony phylogenetic analysis was performed using the software TNT 1.5 (*Goloboff & Catalano, 2016*). The data matrix used herein was updated from *Tong & Li (2019)* and *Mallon & Brinkman (2018)*, and includes 17 taxa and 50 characters. *Adocus* was set as the outgroup following *Tong & Li (2019)*. Because there are five inframarginal scutes on *Jiangxichelys ganzhouensis* (*Tong et al., 2016*), character 37 was modified to: "Inframarginals: (0) five to three pairs; (1) two pairs; (2) absent". In addition, on the basis of *Tong et al. (2016)*, character 48 was changed in *Jiangxichelys ganzhouensis* from ? to 1 (*i.e.,* ratio of midline epiplastral suture length to total midline plastral length greater than 0.1). The length to width ratios of the carapace of *Nanhsiungchelys* and *Anomalochelys* are equal to or larger than 1.6 (*Hirayama et al., 2001*; *Hirayama et al., 2009*; *Tong & Li, 2019*), whereas the other genera (*e.g.,* *Basilemys*) exhibit smaller ratios (*Mallon & Brinkman, 2018*).
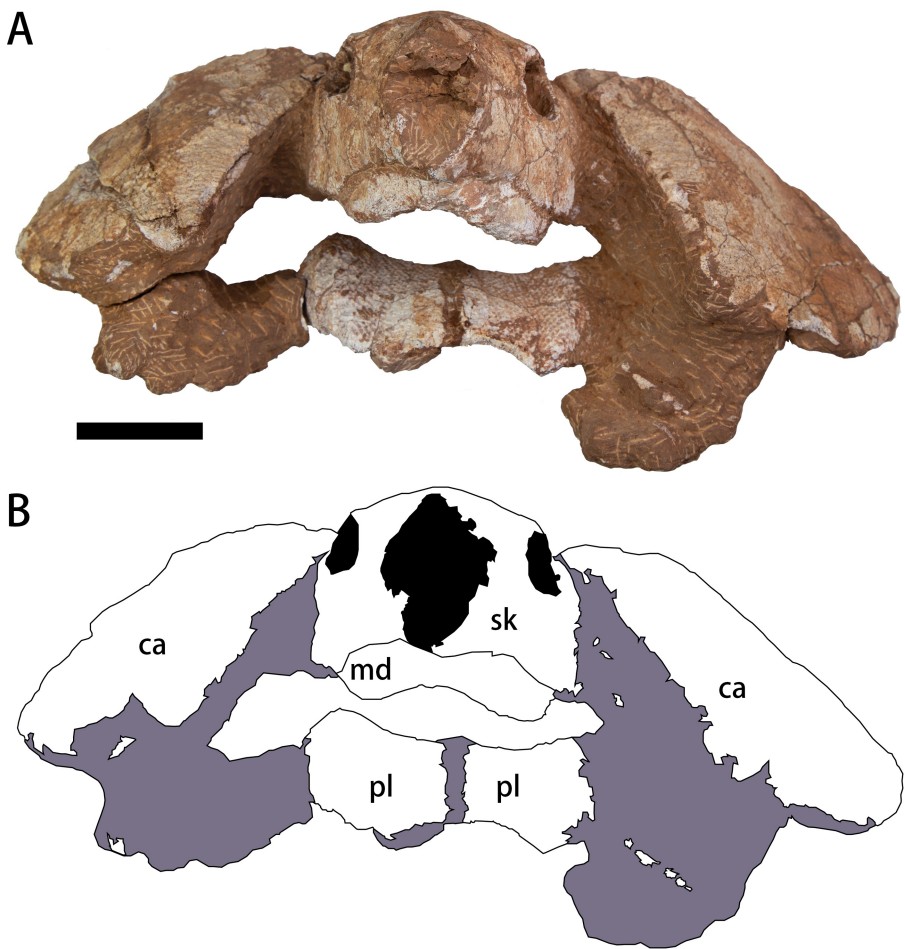

A

B

ca

sk

md

ca

pl    pl

**Figure 2** **Photograph (A) and outline drawing (B) of *Nanhsiungchelys yangi* (CUGW VH108) in anterior view.** Gray and black parts indicate the surrounding rock and openings of the skull, respectively. Scale bar equals five cm. Abbreviations: ca, carapace; md, mandible; pl, plastron; sk, skull.

An example with a ratio between 1.4 and 1.6 has not been found in any nanhsiungchelyids yet. Therefore, a new character was added: "Length to width ratio of the carapace: (0) less than 1.4; (1) equal to or larger than 1.6". Moreover, *Yuchelys nanyangensis* was added to the data matrix based on *Tong et al. (2012)*. A total of 13 characters out of 50 could be coded for *Nanhsiungchelys yangi*, representing only 26% of the total number of characters. This is because the new species is based on a partial specimen missing many of the features scored in other taxa. The analysis was conducted using a traditional search with 1000 replicates. A tree bisection reconnection (TBR) swapping algorithm was employed, and 10 trees were saved per replicate. All characters were treated as unordered and of equal weight. Standard bootstrap support values were calculated using a traditional search with 100 replicates. Bremer support values were also calculated (*Bremer, 1994*). In addition, a time-scaled phylogeny was generated in R (*R Core Team, 2023*) using our strict consensus tree and the first/last appearance datum (FAD/LAD) of all taxa. The R package Strap (*Bell*

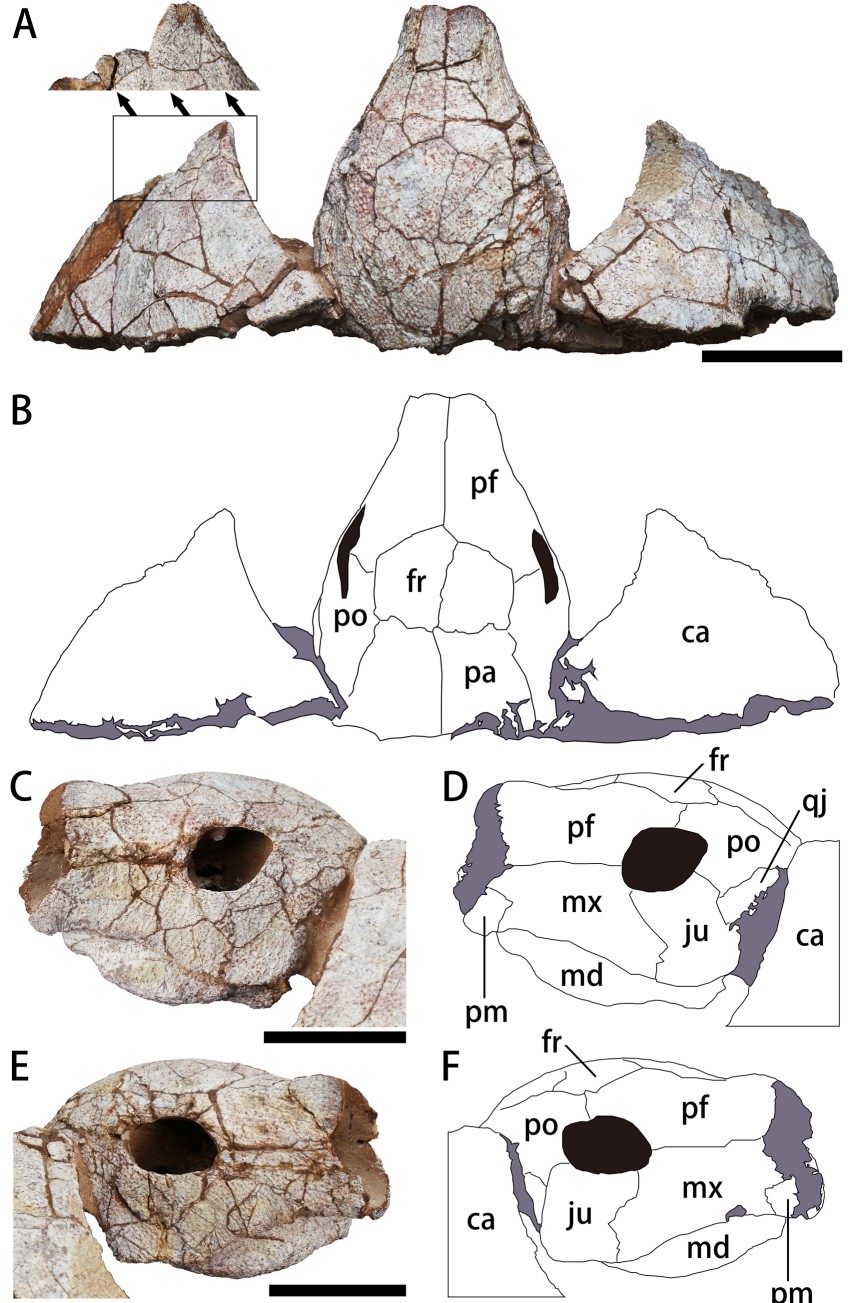

**Figure 3 The skull and carapace of *Nanhsiungchelys yangi* (CUGW VH108).** (A, B) Photograph and outline drawing of the skull and carapace in dorsal view, with a magnified view showing a distinct protrusion at the tip of anterolateral process (perpendicular to the surface of the carapace). (C, D) Photograph and outline drawing of the skull in left lateral view. (E, F) Photograph and outline drawing of the skull in right lateral view. Gray and black parts indicate the surrounding rock and openings of the skull, respectively. Scale bars equal five cm. Abbreviations: ca, carapace; fr, frontal; ju, jugal; md, mandible; mx, maxilla; pa, parietal; pf, prefrontal; pm, premaxilla; po, postorbital; qj, quadratojugal.

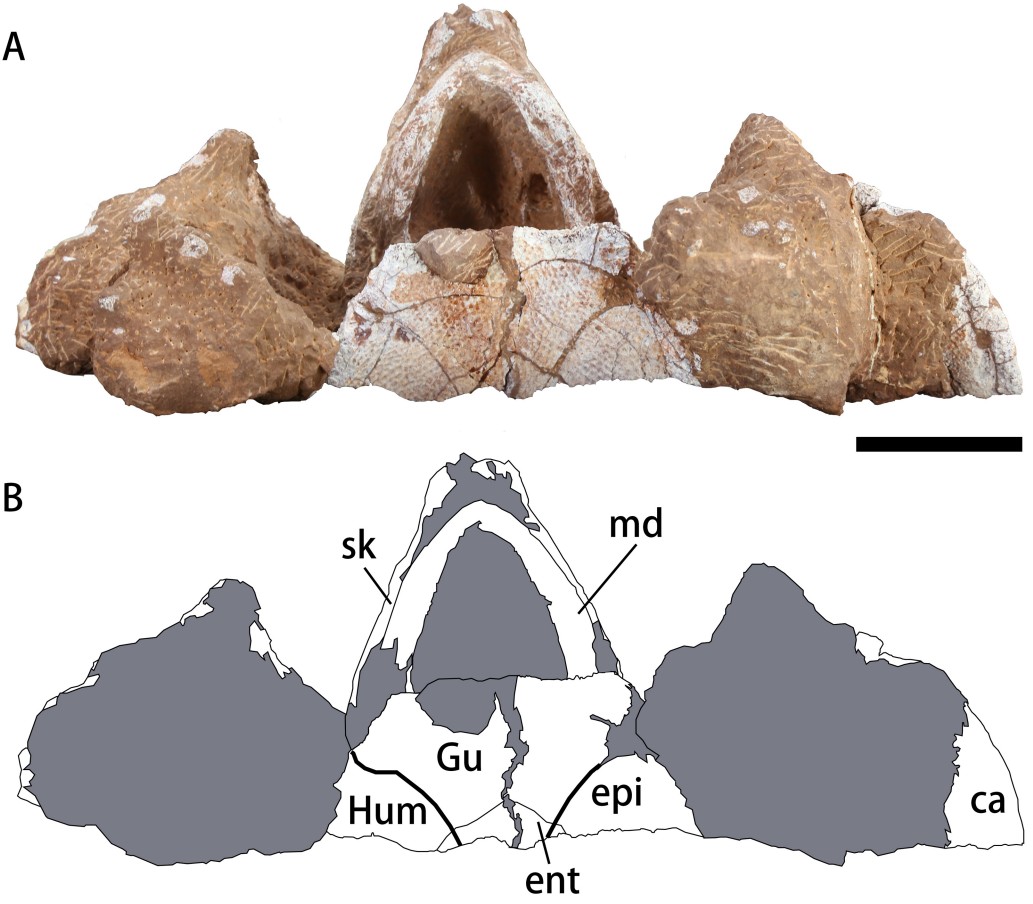

**Figure 4 Photograph (A) and outline drawing (B) of *Nanhsiungchelys yangi* (CUGW VH108) in ventral view.** Bold lines represent the sulci between scutes and gray parts indicate the surrounding rock. Scale bar equals five cm. Abbreviations: ca, carapace; ent, entoplastron; epi, epiplastron; Gu, gular scute; Hum, humeral scute; md, mandible; sk, skull.

*& Lloyd, 2014*) was used to estimate divergence times and the function geoscalePhylo was used to plot the time-scaled tree against a geological timescale.

## Computational fluid dynamics

Computational fluid dynamics (CFD) simulations of water flow were performed in the software COMSOL Multiphysics (*v.* 5.6). Three-dimensional digital models of *Nanhsiungchelys yangi* and two 'hypothetical turtles' without anterolateral processes were created using COMSOL's in-built geometry tools. These models were placed in cylindrical flow domains, with the material properties of water assigned to the space surrounding the models and the swimming speeds of an extant large turtle used as flow velocities at the inlet. CFD simulations were performed using a stationary solver, and based on the results drag forces were extracted for each model. The main steps including the construction of digital models, specification of fluid properties and boundary conditions, meshing, and computation are detailed in Supplemental Information 3.

## RESULTS

### Systematic paleontology

Testudines Linnaeus, 1758
Cryptodira Cope, 1868
Nanhsiungchelyidae *Yeh, 1966*
*Nanhsiungchelys Yeh, 1966*

**Emended diagnosis**. A genus of Nanhsiungchelyidae of medium–large size, with an entire carapace length of 0.5–1.1 m. The surface of the skull, lower jaw, and both carapace and plastron are covered with sculpturing consisting of large pits formed by a network of ridges. Temporal emargination and cheek emargination are shallow; orbits located at about mid-length of the skull and facing laterally; jugal forms the lower margin of the orbit. Carapace elongate, with a deep nuchal emargination and a pair of large anterolateral processes that extend forward and are formed entirely by the first peripheral; wide neural plates and vertebral scutes; gulars fused and extend deeply onto the entoplastron; extragulars absent; complete row of narrow inframarginals. Wide angle between the acromion process and scapula process of about 105°. One large dermal plate located above the manus.
**Type species.** *Nanhsiungchelys wuchingensis Yeh, 1966*
**Distribution.** Guangdong, China

*Nanhsiungchelys yangi* sp. nov.
 urn:lsid:zoobank.org:act:CFF09330-A60B-4239-BAEB-54E856DAD4DC
**Etymology.** The species epithet *yangi* is in memory of paleontologist Zhongjian Yang (Chung-Chien Young).
**Holotype.** CUGW VH108, a partial skeleton comprising a well-preserved skull and lower jaw and the anterior parts of the carapace and plastron (Figs. 2–4).
**Locality and horizon.** Nanxiong, Guangdong, China. Dafeng Formation, Upper Cretaceous, Cenomanian to middle Campanian (*Xi et al., 2021*).
**Diagnosis.** A medium-sized species of *Nanhsiungchelys* with an estimated entire carapace length of more than 0.5 m. It differs from *Nanhsiungchelys wuchingensis* in the following combination of characters: snout is triangular in dorsal view; premaxilla greater in height than length; posteroventral ramus of the maxilla extends to the ventral region of the orbit; the dorsal margin of the maxilla is relatively straight; jugal is greater in height than width; prefrontal is convex dorsally behind the apertura narium externa; temporal emargination is mainly formed by the parietal; paired parietals are bigger than the frontals in dorsal view; middle and posterior parts of the mandible are more robust than the most anterior part in ventral view; anterolateral processes is wide; and the angle between the two anterior edges of the entoplastron is wide (~110°).

## Description
### General aspects of the skull
The skull is large, with a length of 13 cm (Figs. 3A, 3B). It is well preserved, but numerous cracks on its outer surface limit the identification of bone sutures. The snout (*i.e.,* the parts anterior to the orbit) is large, equal to about 1/3 the length of the skull, and longer than in *Jiangxichelys neimongolensis* and *Zangerlia ukhaachelys* (*Joyce & Norell, 2005*; *Brinkman et al., 2015*). In dorsal view, the snout is close to triangular in outline with a narrow anterior end (Figs. 3A, 3B). In lateral views, the robust snout is nearly as deep as the whole skull, with the anterior end roughly perpendicular to the horizon (Figs. 3C–3F). These features differ from *Nanhsiungchelys wuchingensis* in which the snout is flattened, with the anterior end increasing in width in dorsal view (*Tong & Li, 2019*), giving it a trumpet shape. A large apertura narium externa is located in the front part of the snout, which is roughly lozenge shaped and greater in height than width in anterior view (Fig. 2). Because the posterior part of the skull is not preserved, it is difficult to accurately determine the morphological characteristics of cheek emargination (Figs. 3C–3F). Nevertheless, based on the visible bone morphology, we infer that the cheek emargination was absent or low, rather than deep (*i.e.,* to the level or even beyond the level of orbit, see *e.g., Emydura macquarrii*) (*Li & Tong, 2017*). Posteriorly, the temporal emargination is weakly developed (Figs. 3A, 3B), which is similar to *Nanhsiungchelys wuchingensis* (*Tong & Li, 2019*) and the 'Hefei specimen' (*Hu et al., 2016*), but differs from *Jiangxichelys neimongolensis, J. ganzhouensis* and *Zangerlia ukhaachelys* (*Brinkman & Peng, 1996*; *Joyce & Norell, 2005*; *Tong et al., 2016*). The surface of the skull (as well as those of the carapace and plastron) is covered with a network of sculptures consisting of pits and ridges, which is one of the synapomorphies of Nanhsiungchelyidae (*Li & Tong, 2017*).

### Premaxilla
A small bone in the anterior and ventral part of the maxilla is identified as the premaxilla (Figs. 3C–3F). It is greater in height than width, similar to *Jiangxichelys neimongolensis* and *Zangerlia ukhaachelys* (*Joyce & Norell, 2005*; *Brinkman et al., 2015*), but differs from *Nanhsiungchelys wuchingensis* in which the premaxilla is wider than it is high in lateral view and has an inverse Y-shape in ventral view (*Tong & Li, 2019*). Given the existence of the large lozenge-shaped external narial opening, the contact between the left and right premaxillae may be short, unlike the condition of *Jiangxichelys neimongolensis* (*Brinkman et al., 2015*). However, the poor preservation of elements near the external narial opening prevents more detailed observations, and the possibility of a Y-shaped premaxilla as in *Nanhsiungchelys wuchingensis* cannot be excluded.

### Maxilla
The maxilla is large and trapezoidal in outline (Figs. 3C–3F). The main body is located anterior to the orbit, but the posteroventral ramus extends to the ventral region of the orbit, which differs from the situation in *Nanhsiungchelys wuchingensis*, in which the maxilla is located entirely anterior to the orbit (*Tong & Li, 2019*), and also differs from that in most other turtles (including *Zangerlia ukhaachelys* and *Jiangxichelys neimongolensis*), in which

the maxilla contributes to the lower rim of the orbit (*Joyce & Norell, 2005*; *Brinkman et al., 2015*). In lateral view, the dorsal margin of the maxilla is relatively straight and extends posteriorly to the mid-region of the eye socket, which is similar to the condition in some extant turtles (*e.g.*, *Platysternon megacephalum*) (*Li & Tong, 2017*). However, this differs from the condition in *Nanhsiungchelys wuchingensis* in which the top of the maxilla is curved dorsally (*Tong & Li, 2019*), and also differs from *Zangerlia ukhaachelys* and *Jiangxichelys neimongolensis* in which the top of the maxilla tapers anterdorsally (*Joyce & Norell, 2005*; *Brinkman et al., 2015*).

## Jugal

The jugal is shaped like a parallelogram in lateral view (Figs. 3C–3F). It is greater in height than width, unlike *Nanhsiungchelys wuchingensis*, in which the jugal is wider than it is high (*Tong & Li, 2019*). The jugal consists of the lower rim of the orbit, which is similar to that of *Nanhsiungchelys wuchingensis*, but differs from most turtles, in which this structure is mainly formed by the maxilla (*Tong & Li, 2019*). The jugal of *Nanhsiungchelys yangi* also differs from that of *Jiangxichelys ganzhouensis*, in which the jugal is more posteriorly located (*Tong et al., 2016*). The jugal contacts with the maxilla anteriorly, and this suture is sloped. The terminal parts of the jugal contacts with the quadratojugal.

## Quadratojugal

The bone that is posterior to the jugal and ventral to the postorbital is identified as the quadratojugal (Figs. 3C–3F). Its location is similar in *Nanhsiungchelys wuchingensis* (*Tong & Li, 2019*), but the full shape is uncertain due to covering by the carapace.

## Prefrontal

In dorsal view, each prefrontal is large and elongate anteroposteriorly, and narrows anteriorly and enlarges posteriorly (Figs. 3A, 3B). The portion in front of the orbit is entirely composed of the prefrontal (Figs. 3A, 3B), which differs from *Nanhsiungchelys wuchingensis* in which the maxilla extends dorsally to the prefrontal and occupies some space (*Tong & Li, 2019*). The paired prefrontals contact each other at the midline and form an approximate arrow shape. They form the dorsal margin of apertura narium externa anteriorly, the anterodorsal rim of the orbit posterolaterally, and contact the frontal and postorbital posteriorly (Figs. 3A, 3B). The contact area between the prefrontal and frontal is convex anteriorly (*i.e.*, ' Λ'-shaped), which is similar to that seen in *Nanhsiungchelys wuchingensis* (*Tong & Li, 2019*). In lateral view, the prefrontal is anterior to the postorbital and dorsal to the maxilla, and consists of the anterodorsal rims of the orbit (Figs. 3C–3F). This is similar to the anatomy in *Nanhsiungchelys wuchingensis*, *Jiangxichelys neimongolensis* and *Zangerlia ukhaachelys* (*Brinkman & Peng, 1996*; *Joyce & Norell, 2005*; *Tong & Li, 2019*). Behind the apertura narium externa, the prefrontal is convex dorsally (Figs. 3C–3F), rather than concave as in *Nanhsiungchelys wuchingensis* (*Tong & Li, 2019*).

## Frontal

The paired frontals form a large pentagon that is located in the center of the skull roof (Figs. 3A, 3B), which is similar to the condition in *Nanhsiungchelys wuchingensis* and *Zangerlia*

*ukhaachelys* (*Joyce & Norell, 2005*; *Tong & Li, 2019*). In these taxa, the anterior margins constitute a " Λ" shape for articulating with the prefrontal. The lateral and posterior margins contact the postorbital and parietal, respectively. The frontal is excluded from the rim of the orbit, as in *Nanhsiungchelys wuchingensis* and *Zangerlia ukhaachelys* (*Joyce & Norell, 2005*; *Tong & Li, 2019*). Notably, a line between the paired frontals (Figs. 3A, 3B) might be a suture or crack. We think it most likely represents a suture because a similar structure appears in other nanhsiungchelyid specimens (*Joyce & Norell, 2005*; *Tong & Li, 2019*). Interestingly, this suture is unusually slanted, which may be the result of developmental abnormality and needs more specimens for verification.

## Postorbital

The postorbital is subtriangular in outline and elongated anteroposteriorly, and it composes part of the lateral skull roof. Most parts of the postorbital are behind the orbit, but the anterodorsal process extends to the dorsal edge of the orbit (Figs. 3C–3F). Thus, the postorbital consists of the posterior-upper and posterior rims of the orbits, which is similar to the elements of *Nanhsiungchelys wuchingensis*, *Jiangxichelys ganzhouensis* and *Zangerlia ukhaachelys* (*Joyce & Norell, 2005*; *Tong et al., 2016*; *Tong & Li, 2019*). The postorbital contacts the prefrontal and frontal anteriorly, the jugal and quadratojugal ventrally, and the parietal medially (Figs. 3A–3F). In dorsal view, the shape of the posterior margin of the postorbital is uncertain due to its poor preservation and because it is partly obscured by the carapace. It is also uncertain if the postorbital constitutes the rim of temporal emargination. Notably, the postorbital in both *Nanhsiungchelys yangi* and *N. wuchingensis* is relatively large in size (*Tong & Li, 2019*), whereas just a small element forms the 'postorbital bar' in *Jiangxichelys ganzhouensis* and *Zangerlia ukhaachelys* (*Joyce & Norell, 2005*; *Tong et al., 2016*).

## Parietal

The trapezoidal parietal contributes to the posterior part of the skull roof (Figs. 3A, 3B), which is similar to the condition in *Nanhsiungchelys wuchingensis* (*Tong & Li, 2019*). However, the paired parietals are bigger than the frontals in dorsal view, contrasting with the configuration in *Nanhsiungchelys wuchingensis* (*Tong & Li, 2019*). The parietal contacts the frontal anteriorly and the postorbital laterally, and these boundaries are not straight. Posteriorly, the parietal contributes to the upper temporal emarginations, but the absence of the posterior ends of the parietal (especially the right part) hampers the identification of the rims of upper temporal emarginations.

## Mandible

The mandible is preserved *in situ* and tightly closed with the skull (Figs. 3C–3F). The location of the mandible is posterior and interior to the maxillae (Fig. 4). As a result, the beak is hidden, but the lower parts of the mandible can be observed. The symphysis is fused, which is similar to the mandible of *Nanhsiungchelys wuchingensis* (*Tong & Li, 2019*). In ventral view, the most anterior part of the mandible appears slender, but the middle and posterior parts are robust (Fig. 4). This differs from *Nanhsiungchelys wuchingensis*, in which nearly all parts of the mandible are equal in width (*Tong & Li, 2019*).

## Carapace

Only the anterior parts of the carapace are preserved (Figs. 3A, 3B). The preserved parts indicate a deep nuchal emargination and a pair of anterolateral processes, which are similar to those of *Anomalochelys angulata*, *Nanhsiungchelys wuchingensis*, *Nanhsiungchelys* sp. (SNHM 1558), and the 'Hefei specimen' (*Hirayama et al., 2001*; *Hirayama et al., 2009*; *Hu et al., 2016*; *Tong & Li, 2019*). In contrast, the carapaces of other genera of nanhsiungchelyids (including *Basilemys*, *Hanbogdemys*, *Kharakhutulia*, *Jiangxichelys* and *Zangerlia*) usually have a shallow nuchal emargination and/or lack the distinctive anterolateral processes (*Mlynarski, 1972*; *Sukhanov, 2000*; *Sukhanov, Danilov & Syromyatnikova, 2008*; *Tong & Mo, 2010*; *Danilov, Sukhanov & Syromyatnikova, 2013*; *Mallon & Brinkman, 2018*). In dorsal view, each anterolateral process of *Nanhsiungchelys yangi* is very wide (nearly 90°), similar to *Nanhsiungchelys wuchingensis* (*Tong & Li, 2019*); however, the anterolateral processes of *Anomalochelys angulata* and *Nanhsiungchelys* sp. (SNHM 1558) are slender crescent-shaped and horn-shaped, respectively, both of which are sharper than in *Nanhsiungchelys yangi* (*Hirayama et al., 2001*; *Hirayama et al., 2009*). Among the above species of *Nanhsiungchelys* and *Anomalochelys*, there is always a distinct protrusion at the tip of each anterolateral process, and this protrusion becomes more prominent in *Anomalochelys angulata* (Fig. 5B) and *Nanhsiungchelys* sp. (SNHM 1558) (*Hirayama et al., 2001*; *Hirayama et al., 2009*). In *Nanhsiungchelys wuchingensis* and *Anomalochelys angulata* the most anterior end of the process shows varying degrees of bifurcation (Fig. 5B) (*Hirayama et al., 2001*; *Tong & Li, 2019*), but this bifurcation does not occur in *Nanhsiungchelys yangi* and *Nanhsiungchelys* sp. (SNHM 1558) (*Hirayama et al., 2009*). Due to the lack of sutures preserved on the surface of the carapace, it is difficult to determine whether these processes are composed of nuchal or peripheral plates. However, considering the similarity in shape of the anterolateral processes in *Nanhsiungchelys yangi* and *N. wuchingensis*, the anterolateral processes of *N. yangi* may be formed by the first peripheral plates (as in *N. wuchingensis*).

## Plastron

A large plate under the mandible is identified as the anterior part of the plastron (Fig. 4). The anterior edge of the epiplastron extends anteriorly beyond the deepest part of nuchal emargination (Fig. 4), similar to that seen in *Basilemys*, *Hanbogdemys*, *Jiangxichelys*, *Nanhsiungchelys*, and *Zangerlia* (*Sukhanov, 2000*; *Danilov, Sukhanov & Syromyatnikova, 2013*; *Brinkman et al., 2015*; *Tong et al., 2016*; *Mallon & Brinkman, 2018*; *Tong & Li, 2019*). The anterior part of the epiplastron is very thin, but it increases in thickness posteriorly and laterally (Fig. 2). Although poorly preserved, the angle between the left and right edges can be measured as about 55°, which is wider than *Hanbogdemys orientalis* (*Sukhanov, 2000*). The epiplastra are paired and connected at the midline. Because only the anterior part of the entoplastron is preserved, it is hard to discern its shape. The anterior edges of the entoplastron are strongly convex, and lead into the posterior part of the epiplastra. The angle between the two anterior edges (>110°) is larger than in *Nanhsiungchelys wuchingensis* (~100°) (*Tong & Li, 2019*). The only identifiable scutes are the gular and the humeral. In many nanhsiungchelyids, like *Basilemys praeclara*, *B. morrinensis*, *Jiangxichelys*
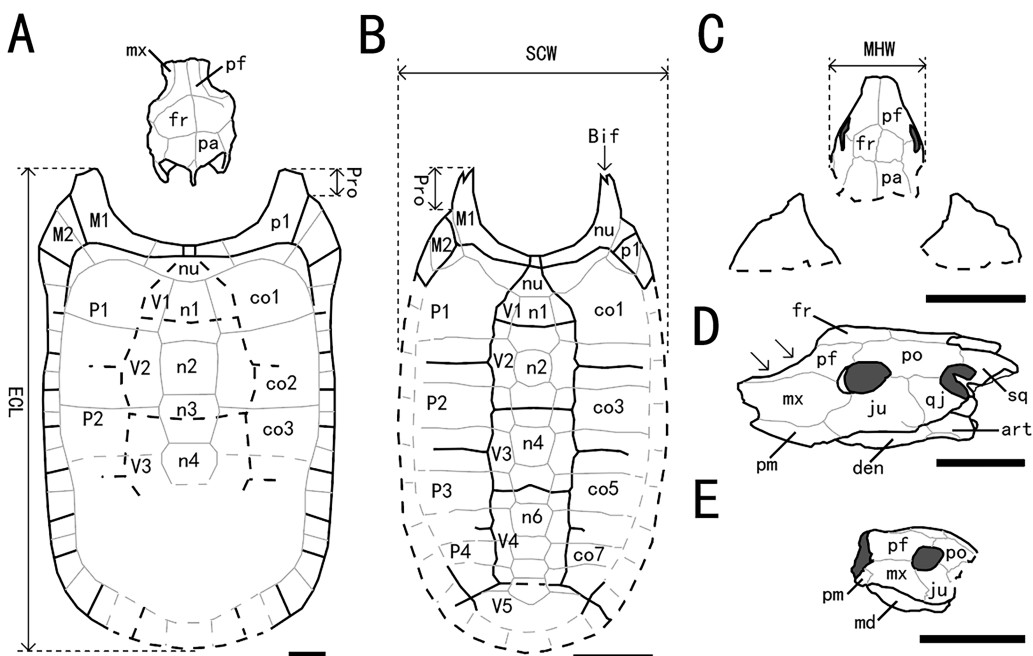

**Figure 5 Outline drawings of three nanhsiungchelyids.** (A) Skull and carapace of *Nanhsiungchelys wuchingensis*, after *Tong & Li (2019)* and *Hirayama et al. (2001)*. (B) Carapace of *Anomalochelys angulata*, after *Hirayama et al. (2001)*. (C) Skull and partial carapace of *Nanhsiungchelys yangi* (CUGW VH108). (D) Skull of *Nanhsiungchelys wuchingensis* in left lateral view, after *Tong & Li (2019)*; arrows indicate the concave prefrontal. (E) Skull of *Nanhsiungchelys yangi* (CUGW VH108) in left lateral view. Scale bars equal 10 cm. Bold black lines represent the sulci between scutes, thin gray lines indicate the sutures between bones, and dashed lines indicate a reconstruction of poorly preserved areas. Abbreviations: bones: art, articular; Bif, bifurcation; co, costal; den, dentary; fr, frontal; ju, jugal; md, mandible; mx, maxilla; n, neural; nu, nuchal; p, peripheral; pa, parietal; pf, prefrontal; pm, premaxilla; po, postorbital; Pro, protrusion; qj, quadratojugal; sq, squamosal; scutes: M, marginal scute; P, pleural scute; V, vertebral scute; measurement: ECL, entire carapace length; MHW, maximum head width; SCW, straightline carapace width.

*ganzhouensis*, *J. neimongolensis*, *Hanbogdemys orientalis*, *Zangerlia dzamynchondi* and *Kharakhutulia kalandadzei* (*Brinkman & Nicholls, 1993*; *Brinkman & Peng, 1996*; *Sukhanov, 2000*; *Sukhanov, Danilov & Syromyatnikova, 2008*; *Danilov, Sukhanov & Syromyatnikova, 2013*; *Tong et al., 2016*; *Mallon & Brinkman, 2018*), extragular scutes usually occur beside the gular scutes, but this does not occur in *Nanhsiungchelys wuchingensis* (*Tong & Li, 2019*) and *N. yangi*. Moreover, the location and shape of the sulci of *Nanhsiungchelys yangi* are similar to those seen in *N. wuchingensis* (*Tong & Li, 2019*). In *Nanhsiungchelys yangi*, the sulcus between the gular and humeral scutes can be identified, and it is slightly curved and extend onto the entoplastron, which is similar to the configuration seen in *Jiangxichelys neimongolensis* and *Nanhsiungchelys wuchingensis* (*Brinkman & Peng, 1996*; *Brinkman et al., 2015*; *Tong & Li, 2019*). However, in the other nanhsiungchelyids (*e.g.*, *Kharakhutulia kalandadzei*, *Zangerlia dzamynchondi*, *Hanbogdemys orientalis*, *Yuchelys nanyangensis* and *Jiangxichelys ganzhouensis*), this sulcus is tangential to (or separated from) the entoplastron

(*Sukhanov, 2000*; *Sukhanov, Danilov & Syromyatnikova, 2008*; *Tong et al., 2012*; *Danilov, Sukhanov & Syromyatnikova, 2013*; *Tong et al., 2016*).

## DISCUSSION

### Taxonomy

Through comparison with a complete specimen (IVPP V3106) of *Nanhsiungchelys wuchingensis*, the large skull (length = 13 cm) of CUGW VH108 is inferred to correspond to an entire carapace length of ~55.5 cm (see Fig. 5A for a definition of 'entire carapace length', which comes from *Hirayama et al. (2001)*). This large body size, coupled with the network of sculptures on the surface of the skull and shell, clearly demonstrates that CUGW VH108 belongs to Nanhsiungchelyidae (*Li & Tong, 2017*). Moreover, CUGW VH108 has a laterally thickened epiplastron (Fig. 2), with the anterior edge of the epiplastron extending anterior of the deepest part of nuchal emargination (Fig. 4), additional features that are diagnostic of Nanhsiungchelyidae (*Li & Tong, 2017*).

Within Nanhsiungchelyidae, CUGW VH108 differs from *Basilemys*, *Hanbogdemys*, *Kharakhutulia*, *Yuchelys*, and *Zangerlia* because all of these taxa have weak nuchal emargination and/or lack distinct anterolateral processes (*Mlynarski, 1972*; *Sukhanov, 2000*; *Sukhanov, Danilov & Syromyatnikova, 2008*; *Tong et al., 2012*; *Danilov, Sukhanov & Syromyatnikova, 2013*; *Mallon & Brinkman, 2018*). Moreover, CUGW VH108 differs from *Jiangxichelys ganzhouensis* and *J. neimongolensis* in which the cheek emargination and temporal emargination are deep (*Brinkman & Peng, 1996*; *Tong et al., 2016*). Although the carapace of both *Anomalochelys* and CUGW VH108 have deep nuchal emargination and a pair of anterolateral processes, the former's anterolateral processes are slender crescent-shaped and have a bifurcated anterior end (*Hirayama et al., 2001*), which are clear differences from the wide processes of CUGW VH108.

CUGW VH108 is assigned to the genus *Nanhsiungchelys* based on the deep nuchal emargination, pair of anterolateral processes, and weakly developed cheek emargination and temporal emargination (*Li & Tong, 2017*). However, CUGW VH108 differs from *Nanhsiungchelys wuchingensis* in which the snout is trumpet shaped (*Tong & Li, 2019*). Moreover, *Nanhsiungchelys wuchingensis* and CUGW VH108 show some differences in their skeletal features (Table 2). In CUGW VH108 these include: the premaxilla is very small and higher than it is wide (Figs. 3C–3F); the top of the maxilla is straight (in lateral views) (Figs. 3C–3F); the maxilla does not occupy the space of the prefrontal (in dorsal views) (Figs. 3A, 3B); a small portion of the maxilla extends posterior and ventral of the orbit (Figs. 3C–3F); the parallelogram-shaped jugal is greater in height than width (Figs. 3C–3F); the prefrontal is convex dorsally behind the apertura narium externa; the parietals are bigger than the frontals (Figs. 3A, 3B); the middle and posterior parts of the mandible are more robust than the most anterior part in ventral view; and the angle between the two anterior edges of the entoplastron is wide (~110°). It is possible that the snout of the only known specimen of *Nanhsiungchelys wuchingensis* (IVPP V3106) was deformed during the burial process, because its trumpet-shaped morphology has not been reported in any other turtles. However, the post-cranial skeleton does not show much evidence of post-mortem

**Table 2 Main differences among the three species of *Nanhsiungchelys*.**

| Character | *Nanhsiungchelys yangi* | *N. wuchingensis* | *Nanhsiungchelys* sp. (SNHM 1558) |
|---|---|---|---|
| Snout | triangular (in dorsal view) | trumpet shaped | unknown |
| Premaxilla | higher than wide | wider than high in lateral view and has an inverse Y-shape in ventral view | unknown |
| Maxilla | unseen in dorsal views; a small portion of the maxilla extends posterior and ventral of the orbit | visible in dorsal views; the maxilla is located entirely anterior to the orbit | unknown |
| Jugal | higher than wide | wider than high | unknown |
| Prefrontal | convex dorsally behind the naris | concave behind the naris | unknown |
| Parietal | bigger than the frontal | smaller than the frontal | unknown |
| Mandible | the middle and posterior parts of the mandible are more robust than the most anterior part in ventral view | nearly all parts of the mandible are equal in width | unknown |
| Entoplastron | the angle between the two anterior edges of the entoplastron is wide (∼110°) | the angle between the two anterior edges of the entoplastron is only ∼100° | unknown |
| Anterolateral processes | wide | wide | slender |
| References | This article | *Tong & Li (2019)* | *Hirayama et al. (2009)* |

deformation, and both *Yeh (1966)* and *Tong & Li (2019)* regarded the unique snout as an original, diagnostic characteristic. CUGW VH108 also differs from *Nanhsiungchelys* sp. (SNHM 1558) in which the anterolateral processes are slender horn-shaped (*Hirayama et al., 2009*). The anterior processes of the 'Hefei specimen' are believed to be long and similar to those of *Anomalochelys angulata* (*Hu et al., 2016*), whereas these are relatively short in CUGW VH108. Thus, CUGW VH108 differs from all other known species of Nanhsiungchelyidae, and herein we erect the new species *Nanhsiungchelys yangi*. Lastly, on the basis of *Tong & Li (2019)* and our new specimen, we emended the diagnosis of *Nanhsiungchelys*. Characteristics shared by both *Nanhsiungchelys wuchingensis* and *N. yangi* are retained, and we exclude the characters that do not match *N. yangi*, such as a long and trumpet-shaped snout, large frontal, and relatively small parietal. This revised diagnosis is listed above.

The differences between *Nanhsiungchelys yangi* and *N. wuchingensis* are not likely to represent ontogenetic variation. Despite only corresponding to half the length of *Nanhsiungchelys wuchingensis* (IVPP V3106), the entire carapace length (∼55.5 cm) of *N. yangi* (CUGW VH108) is still in the middle of the size range reported among Nanhsiungchelyidae. For instance, the entire carapace length of the Chinese nanhsiungchelyid *Jiangxichelys ganzhouensis* is ∼46–74 cm (*Tong et al., 2016*), and the estimated entire carapace length of adult nanhsiungchelyid *Kharakhutulia kalandadzei* is only ∼23–25 cm (*Sukhanov, Danilov & Syromyatnikova, 2008*). In addition, juveniles usually have a larger skull relative to their carapace, whereas mature individuals may have a relatively smaller skull (*Brinkman et al., 2013*). The ratios of maximum head width (MHW) to straightline carapace width (SCW) are ∼30% in both *Nanhsiungchelys yangi* (CUGW VH108) and *N. wuchingensis* (IVPP V3106) (*Tong & Li, 2019*).

Sexual dimorphism is another possible explanation of the observed differences between *Nanhsiungchelys yangi* and *N. wuchingensis*, but this is very difficult to assess. *Cadena et al. (2020)* suggested that horns (similar to the anterolateral processes in *Nanhsiungchelys*) could be used to identify sex in the turtle *Stupendemys geographicus*. However, all known specimens of *Nanhsiungchelys* exhibits distinct anterolateral processes. Some extant male tortoises (*e.g.*, *Centrochelys sulcata*) have a more robust epiplastron than females (*Zhou & Zhou, 2020*), but such a difference has not been reported in *Nanhsiungchelys*. Other lines of evidence (*e.g.*, concavity of the plastron and shape of the xiphiplastral region) commonly used to determine the sex of extant turtles (*Pritchard, 2007*) are also unavailable due to the poor preservation of the above specimens. Based on the above discussion, the most reasonable conclusion is that CUGW VH108 represents a distinct species, rather than the product of intraspecific variation.

**Phylogenetic position and paleobiogeography**
The phylogenetic analysis retrieved seven most parsimonious trees with a length of 77 steps, a consistency index (CI) of 0.675, and a retention index (RI) of 0.679. The strict consensus tree (Fig. 6) recovers *Nanhsiungchelys yangi* and *N. wuchingensis* as sister taxa, with one unambiguous synapomorphy identified: the absence of the extragulars. These two species and *Anomalochelys angulata* form a monophyletic group, which is consistent with the results of *Tong & Li (2019)*. Synapomorphies of this group include wide neurals, first vertebral scute with lateral edges converging anteriorly, cervical scute as wide as long, and the length to width ratio of the carapace is larger than 1.6. In particular, our new character (character 50, the length to width ratio of the carapace) supports this relationship, suggesting it could prove informative in other studies of turtle phylogeny. However, the standard bootstrap and Bremer supports values are low among these groups, and their relationships therefore need further consideration. Interestingly, our new results identify *Yuchelys nanyangensis* and *Zangerlia testudinimorpha* as sister taxa, and this relationship was supported by one unambiguous synapomorphy (their fifth vertebral almost fully covers the suprapygal). However, this relationship needs to be tested in future work because the only known specimen of *Yuchelys nanyangensis* (HGM NR09-11-14) is poorly preserved (*Tong et al., 2012*) and only 15 characters could be used in our phylogenetic analysis.

Although *Anomalochelys* and *Nanhsiungchelys* were in similar stages (Fig. 6), they appear to have lived in different regions (southern China and Japan, respectively). In fact, Cretaceous turtle communities in Japan and the rest of Asia (especially China and Mongolia) are closely comparable, with both areas containing representatives of Adocusia, Lindholmemydidae, Sinochelyidae, and Sinemydidae (*Hirayama, Brinkman & Danilov, 2000*). Similar extinct organisms in these regions also include the plant *Neozamites* (*Sun et al., 1993*; *Duan, 2005*), the bivalve *Trigonioides* (*Ma, 1994*; *Komatsu et al., 2007*), and the dinosaur Hadrosaurinae (*Kobayashi et al., 2019*; *Zhang et al., 2020*). *Sun & Yang (2010)* inferred that the Japan Sea did not exist during the Jurassic and Cretaceous, with the Japan archipelago still closely linked to the eastern continental margin of East Asia. This view is also supported by geological and geophysical evidence (*Kaneoka et al., 1990*; *Liu, Wei & Shi, 2017*). In addition to *Anomalochelys angulata* from Hokkaido (*Hirayama et al., 2001*),

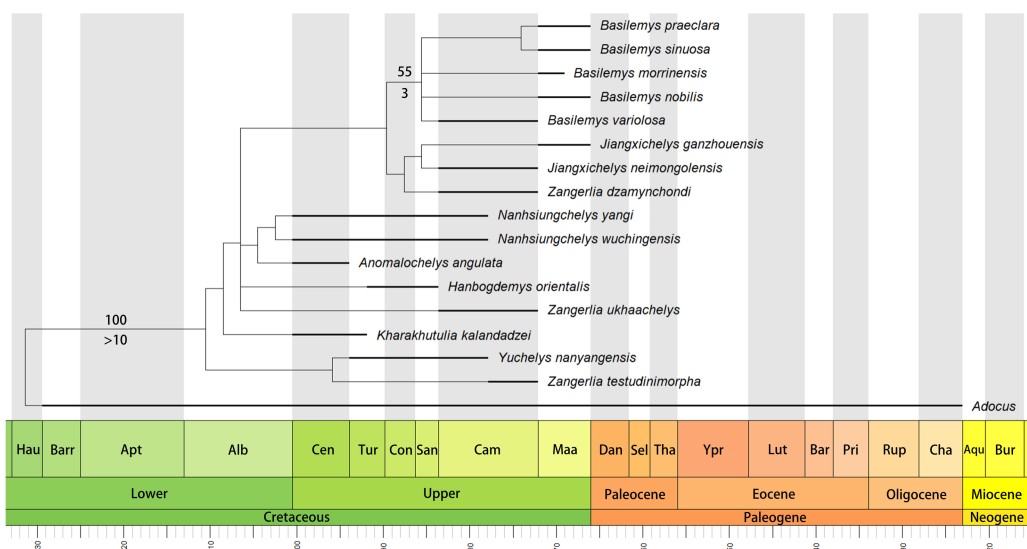

**Figure 6** **Time-scaled strict consensus tree of Nanhsiungchelyidae.** Numbers above nodes are bootstrap support values and numbers below nodes are Bremer support values. Please note that the bootstrap support values less than 50 and the Bremer support values equal to 1 are not shown here. Temporal distributions of species based on *Danilov, Sukhanov & Syromyatnikova (2013)*, *Li & Tong (2017)*, *Syromyatnikova & Danilov (2009)*, *Tong et al. (2016)*, *Mallon & Brinkman (2018)*, and *Xi et al. (2021)*. Abbreviations: Hau, Hauterivian; Barr, Barremian; Apt, Aptian; Alb, Albian; Cen, Cenomanian; Tur, Turonian; Con, Coniacian; San, Santonian; Cam, Campanian; Maa, Maastrichtian; Dan, Danian; Sel, Selandian; Tha, Thanetian; Ypr, Ypresian; Lut, Lutetian; Bar, Bartonian; Pri, Priabonian; Rup, Rupelian; Cha, Chattian; Aqu, Aquitanian; Bur, Burdigalian.

many fragments of Nanhsiungchelyidae (as *Basilemys* sp.) have also been found on Honshu and Kyushu islands, Japan (*Hirayama, 1998*; *Hirayama, 2002*; *Danilov & Syromyatnikova, 2008*). In China, the easternmost specimen of a nanhsiungchelyid turtle (a fragment of the shell) was recovered from the Upper Cretaceous of Laiyang, Shandong (*Li & Tong, 2017*), which is near the west coast of the Pacific Ocean and close to Japan geographically. This geographical proximity likely allowed nanhsiungchelyids to disperse between China and Japan during the Late Cretaceous.

## Function of the anterolateral processes of the carapace

The anterolateral processes of *Nanhsiungchelys* (and *Anomalochelys*) probably performed a variety of functions, but the principal function was most likely self-protection. In the earliest research on *Nanhsiungchelys wuchingensis*, *Yeh (1966)* did not discuss the function of the anterolateral processes, but speculated that the neck was flexible, and the skull could be withdrawn into the shell to avoid danger. This hypothesis was supported by a complete specimen (93NMBY-2) of the nanhsiungchelyid *Jiangxichelys neimongolensis* whose head was withdrawn into the shell (*Brinkman et al., 2015*). In contrast, *Hirayama et al. (2001)* suggested that the large skull could not be fully withdrawn within the shell (parallel to the extant big-headed turtle *Platysternon megacephalum*) and that the anterolateral processes of *Nanhsiungchelys wuchingensis* and *Anomalochelys angulata* were used for protecting the skull. *Hirayama et al. (2001)* also noted that *Nanhsiungchelys* has undeveloped temporal

emargination, whereas *Jiangxichelys* has distinct temporal emargination, and the former condition could inhibit the ability to retract the skull inside the shell (*Hirayama et al., 2009*; *Werneburg, 2015*; *Hermanson et al., 2022*). Together, this suggests that despite the possession of a flexible neck that could have made it possible to retract the head, the large size of the skull and the reduced temporal emargination were considerable obstacles to doing so. Today, turtles that cannot retract the head are restricted to a few aquatic groups (*e.g.*, Platysternidae) (*Zhou & Li, 2013*), whereas most turtles (including all tortoises) have this capability (*Zhou & Zhou, 2020*). An additional strong piece of evidence that *Nanhsiungchelys* could not retract the head is that the skulls of all known specimens (IVPP V3106, SNHM 1558, and CUGW VH108) are preserved outside of the shell, and the anterolateral processes would thus provide lateral protection for the head (*Yeh, 1966*; *Hirayama et al., 2009*; *Tong & Li, 2019*). Nevertheless, it seems evident that this protective strategy of *Nanhsiungchelys* was inefficient, because the dorsal side of the head would be left vulnerable to attack, and this may explain why extant terrestrial turtles usually abandon this mode of protection.

The anterolateral processes might also have been used during fighting for mates, as hypothesized for the extinct side-necked turtle *Stupendemys geographicus* (*Cadena et al., 2020*). Nanhsiungchelyids and extant tortoises share many comparable skeletal characteristics (*Hutchison & Archibald, 1986*) and inferred reproductive behaviors (*Ke et al., 2021*), and thus *Nanhsiungchelys* might have been characterized by similar combat behavior. A parallel hypothesis was proposed by *Hirayama & Sonoda (2012)* that the combinations of cranial and nuchal morphology in *Nanhsiungchelys* and *Anomalochelys* could facilitate sexual displays, similar to some extant testudinids. However, all known specimens of *Nanhsiungchelys* and *Anomalochelys* possess distinct anterolateral processes and deep nuchal emargination, suggesting these structures might also have been present in females (although this is uncertain because it is not possible to determine their sex). If so, the anterolateral processes would not be the result of sexual dimorphism and associated combat or display. Another piece of evidence arguing against the fighting view is that there are no scars on the anterolateral processes of CUGW VH108, as might be expected if they were used in fighting.

The anterolateral processes of *Nanhsiungchelys* might also have had a secondary function in reducing drag as the animal was moving through water. Today, some tortoises living in dry areas (*e.g.*, *Aldabrachelys gigantea* and *Centrochelys sulcata*) will immerse themselves in mud or water for a long time to avoid the heat (*Zhou & Zhou, 2020*), and *Aldabrachelys gigantea* could even swim (or float) in the ocean (*Gerlach, Muir & Richmond, 2006*; *Hansen et al., 2016*). Nanxiong Basin was extremely hot (∼27–34 °C) during the Late Cretaceous (*Yang et al., 1993*), and the appearance of diverse fossils of Gastropoda, Bivalvia, Charophyceae, and Ostracoda (*Zhang et al., 2013*) suggests the existence of lakes or rivers. Thus, *Nanhsiungchelys* may have had a parallel lifestyle to these tortoises, and the reduction of drag could have been important under these circumstances. *Nessov (1984)* also mentioned that nanhsiungchelyids would anchor themselves on the bottom of streams to offset drift, which could be an adaptation to strong currents. The anterolateral processes of *Nanhsiungchelys* could have played a role in reducing resistance to fluid motion, and

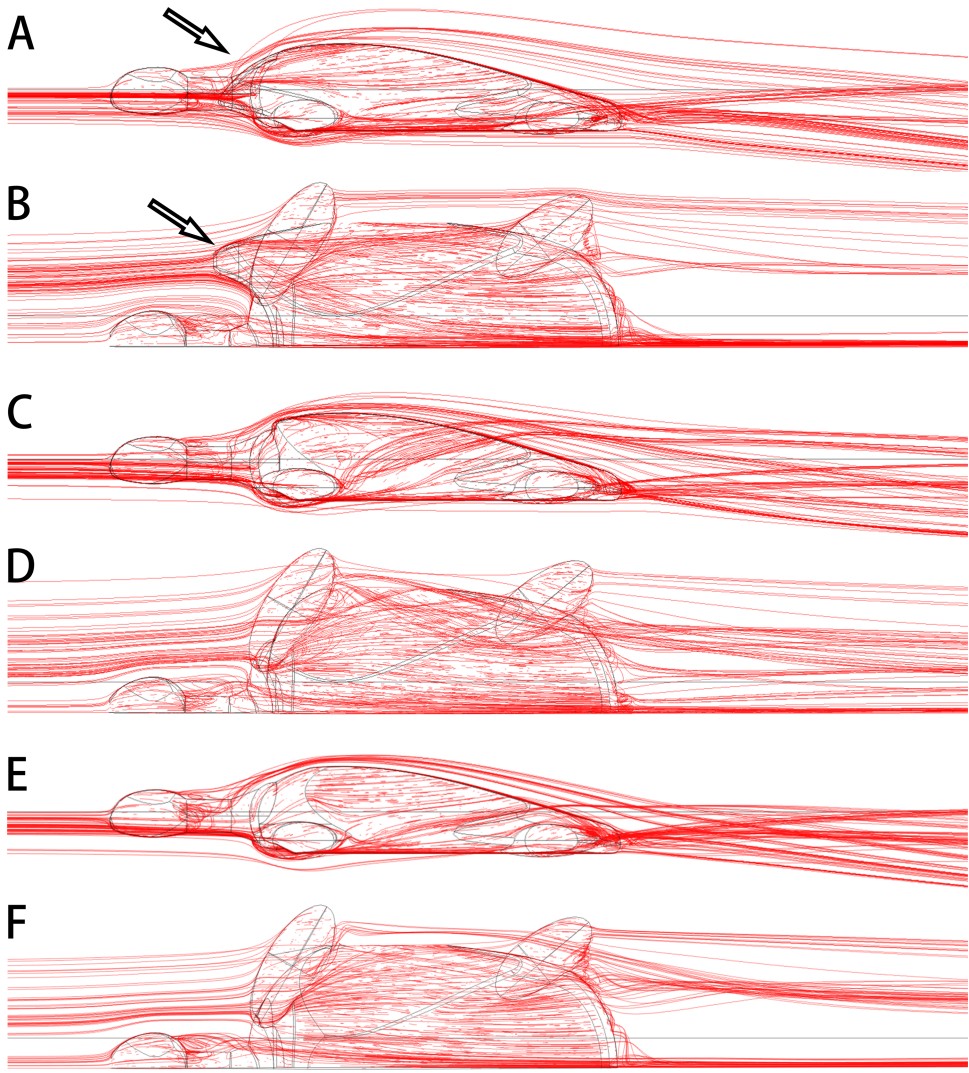

**Figure 7** **3-D plots of streamlines at flow velocities of 1.0 m s⁻¹.** (A & B) model of *Nanhsiungchelys yangi* (in left lateral and dorsal views, respectively); (C & D) model of hypothetical turtle I (in left lateral and dorsal views, respectively), whose anterior carapace and body are blunt; (E & F) model of hypothetical turtle II (in left lateral and dorsal views, respectively), whose anterior carapace is streamlined and similar to most freshwater turtles. The arrows indicate the anterolateral processes. The direction of ambient flow is from left to right.

the efficiency of this would have been close to the level of extant freshwater turtles (see Supplemental Information 3 for detailed information on hydrodynamic analyses). The reason for this is that these processes made the anterior part of the shell more streamlined (Figs. 7A, 7B), analogous to the streamlined fairing on the anterior of airplanes and rockets. However, we acknowledge this remains a hypothesis at this time, because there is no conclusive evidence of swimming in *Nanhsiungchelys*.

Many of the specialized morphological features of nanhsiungchelyids (*e.g.*, huge skull, distinct anterolateral processes, and unusually thick eggshells) are most likely adaptations

to their environment. *Nanhsiungchelys* was a successful genus because it belongs to the only group of turtles that has been reported from the Dafeng Formation, suggesting these unusual turtles were well adapted to their environment. However, their specialist survival strategies might have been very inefficient, because the anterolateral processes could not protect the dorsal side of the head, and the thick eggshell (*Ke et al., 2021*) might have hindered the breathing and hatching of young. All of these features are not present in extant turtles, suggesting this was not a dominant direction in turtle evolution. Consistent with this, nanhsiungchelyids became extinct at the end of the Cretaceous, but many contemporary turtles (*e.g.*, Adocidae, Lindholmemydidae, and Trionychidae) survived into the Cenozoic (*Lichtig & Lucas, 2016*).

## CONCLUSIONS

A turtle skeleton (CUGW VH108) with a well-preserved skull and lower jaw, together with the anterior parts of the shell, was found in Nanxiong Basin, China. This is assigned to the genus *Nanhsiungchelys* based on the large estimated body size (∼55.5 cm), the presence of a network of sculptures on the surface of the skull and shell, shallow cheek emargination and temporal emargination, deep nuchal emargination, and a pair of anterolateral processes on the carapace. Based on the character combination of a triangular-shaped snout (in dorsal view) and wide anterolateral processes, we erect a new species *Nanhsiungchelys yangi*. A phylogenetic analysis of nanhsiungchelyids places *Nanhsiungchelys yangi* and *N. wuchingensis* as sister taxa. We agree with previous suggestions that the anterolateral processes on the carapace could have protected the head, but also infer a potential secondary function for reducing drag force during movement through water. These unique characteristics might have helped nanhsiungchelyids survive in a harsh environment, but did not save them from extinction during the K-Pg event.

**Institutional abbreviations**

| | |
|---|---|
| **CUGW** | China University of Geosciences (Wuhan), Wuhan, China |
| **HGM** | Henan Geological Museum, Zhengzhou, China |
| **IMM** | Inner Mongolia Museum, Huhhot, China |
| **IVPP** | Institute of Vertebrate Paleontology and Paleoanthropology, Chinese Academy of Sciences, Beijing, China |
| **LJU** | Lanzhou Jiaotong University, Lanzhou, China |
| **NHMG** | Natural History Museum of Guangxi, Nanning, China |
| **NMBY** | Nei Mongo Bowuguan, Huhhot, China |
| **SNHM** | Shanghai Natural History Museum, Shanghai, China |
| **UB** | University of Bristol, Bristol, UK |
| **UPC** | China University of Petroleum (East China), Qingdao, China |
| **YSNHM** | Yingliang Stone Natural History Museum, Nan'an, China |

## ACKNOWLEDGEMENTS

We thank Xing Xu (IVPP) for his useful suggestions, thank Kaifeng Wu (YSNHM) for preparing turtle skeleton, and thank Mingbo Wang (UPC), Zichuan Qin (UB), Wen Deng (CUGW) and Haoran Sun (LJU) for assistance with CFD.

### Funding

This work was supported by the National Natural Science Foundation of China (42288201), Guangdong Special Fund for National Park Construction (2021GJGY026). Imran Rahman was supported by joint funding from the US National Science Foundation (NSF-NERC EAR-2007928) and the UK Natural Environment Research Council (NE/V010859/2). The funders had no role in study design, data collection and analysis, decision to publish, or preparation of the manuscript.

### Grant Disclosures

The following grant information was disclosed by the authors:
National Natural Science Foundation of China: 42288201.
Guangdong Special Fund for National Park Construction: 2021GJGY026.
US National Science Foundation: NSF-NERC EAR-2007928.
UK Natural Environment Research Council: NE/V010859/2.

### Competing Interests

The authors declare there are no competing interests.

### Author Contributions

- Yuzheng Ke conceived and designed the experiments, performed the experiments, analyzed the data, prepared figures and/or tables, authored or reviewed drafts of the article, and approved the final draft.
- Imran A. Rahman conceived and designed the experiments, analyzed the data, authored or reviewed drafts of the article, and approved the final draft.
- Hanchen Song conceived and designed the experiments, performed the experiments, analyzed the data, prepared figures and/or tables, and approved the final draft.
- Jinfeng Hu conceived and designed the experiments, performed the experiments, analyzed the data, prepared figures and/or tables, and approved the final draft.
- Kecheng Niu performed the experiments, authored or reviewed drafts of the article, and approved the final draft.
- Fasheng Lou conceived and designed the experiments, authored or reviewed drafts of the article, and approved the final draft.
- Hongwei Li conceived and designed the experiments, authored or reviewed drafts of the article, and approved the final draft.
- Fenglu Han conceived and designed the experiments, performed the experiments, analyzed the data, authored or reviewed drafts of the article, and approved the final draft.

### Data Availability

The raw data are available in Supplemental Information 1 and Supplemental Information 2. These include the taxon-character matrix and the three-dimensional digital models of *Nanhsiungchelys yangi* and two hypothetical turtles.

### New Species Registration

The following information was supplied regarding the registration of a newly described species:

*Nanhsiungchelys yangi* sp. nov.: urn:lsid:zoobank.org:act:CFF09330-A60B-4239-BAEB-54E856DAD4DC

Publication LSID: urn:lsid:zoobank.org:pub:F53B5FA5-D018-453D-814D-C854810EFEFE

### Supplemental Information

Supplemental information for this article can be found online at http://dx.doi.org/10.7717/peerj.15439#supplemental-information.

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
