# Peer review of "A new species of Nanhsiungchelys (Testudines: Cryptodira: Nanhsiungchelyidae) from the Upper Cretaceous of Nanxiong Basin, China"

_PeerJ, doi:10.7717/peerj.15439_

## Round 0.1 · original submission · Minor Revisions

The reviewers are quite positive about this manuscript overall, suggesting just a handful of issues to address. Most of these are requests for additional (minor) clarifications or grammatical corrections, and I am hopeful that they can be incorporated relatively easily.

For the phylogenetic analysis, was Adocus set as the outgroup? I didn't see any mention of outgroup settings in the analysis, but this should be clarified.

I look forward to reading your revised version when it is ready.

·

Basic reporting

see below

Experimental design

see below

Validity of the findings

see below

Additional comments

Dear Editors, Dear Authors,

this manuscript is concerned with the description of new nanhsiungchelyid material from the Late Cretaceous of China.

I already reviewed a previous version of this manuscript for another journal earlier this year. I back then noted some serious flaws to the study as presented: I doubted that the frontals were fused, I found it preposterous to test the swimming abilities of a terrestrial turtle using hydrodynamics, and I doubted the validity of the newly established species. I also highlighted some misunderstandings regarding turtle evolution and literature that the authors had overlooked.

That being said, I am deeply impressed by the revised version I read today: instead of submitting their article to another journal without changes, the authors took my insights into consideration and reworked the manuscript. The most significant change is that the manuscript has been complete re-framed to pertain to a new species of nanhsiungchelyid with discussion of the utility of its unusual anterolateral carapacial processes, instead of the hydrodynamic modeling of a purportedly aquatic turtle. This is very satisfying to me as reviewing manuscripts demands time and as my criticisms were meant to be constructive.

The manuscript is well written and cleanly formatted, so it was a breeze to read. I still am not convinced that the new specimen represents a new species, but as taxonomy is not an exact science, I am willing to let this point go. I otherwise marked up the manuscript with some extremely minor comments that should take minutes to address. The only major point pertains to the type locality of the new taxon, which appears not to be known, either because the authors do not wish to divulge more information or because the specimen was purchased from collectors. I still think it would be a good idea to provide more details.

Best regards,

Walter Joyce

·

Basic reporting

The article meet all the mentioned standards, except that it is missing some relevant references on the topic.

Experimental design

No comment

Validity of the findings

The result of the phylogenetic analysis should be better described and checked given an unexpected basal position of the Zangerlia testudinimorpha + Yuchelys clade.

Additional comments

All comments are included in the attached file

·

Basic reporting

no comment.

Experimental design

no comment

Validity of the findings

no comment

Additional comments

This is a significant contribution to the study of turtles in general and the study of turtles from China in particular. The description of the skull is well written and require no editing that I can see. The argument that the specimens represents a distinct species rather than an ontogenetic stage or sexual dimorph is convincing. The used of computational fluid dynamics to interpret the biology of the animal is innovative and I think will attract considerable interest. I know of a number of turtle taxa that would be good subjects for this kind of study, so I anticipate that future researchers will learn from this study and will incorporate this approach in their work.

---

## Round 0.2 · Minor Revisions

Thank you for your close attention to the reviewer suggestions. In terms of scientific content, the manuscript is ready to go. I have done a final read of the manuscript, and have only a selection of minor stylistic/grammatical edits. Once these are incorporated, I should be able to issue a final acceptance quite quickly.

---

## Round 0.3 · accepted · Accept

Thank you for addressing those final comments. Your manuscript is ready to move forward to publication.